# What Matters In The Structured Pruning of Generative Language Models?

## Abstract

Auto-regressive large language models such as GPT-3 require enormous computational resources to use, leading to huge financial cost and environmental impact. Structured pruning methods traditionally reduce resource usage, however, their application to and efficacy for generative language models is heavily under-explored. We analyze the effects of magnitude, random, and movement (Lagunas et al., 2021) pruning on MLP layers in GPT-like models. We find that movement can underperform for these models while random pruning nearly matches the best methods. By examining neuron-level redundancy measures, we discover that movement does not select neurons based on how *unique* they are compared to other neurons, leaving behind excess redundancy. In view of this, we introduce **G**lobally **U**nique **M**ovement (GUM) to select neurons based on both uniqueness and sensitivity. We then discuss the roles of our techniques on different redundancy metrics through careful comparisons and ablations.

## 1 Introduction

Large language models (LLMs), such as the state-of-the-art GPT-3 model (Brown et al., 2020) with up to 175 billion parameters, have achieved remarkable performance in natural language processing (NLP) tasks. However, training and deploying such massive models also poses significant challenges in terms of computational cost, energy consumption, and environmental impact. Therefore, it is crucial to develop effective methods to reduce the size of LLMs without compromising their quality.

Neural network pruning is a long-standing model compression method (Janowsky, 1989; Mozer & Smolensky, 1988; Frankle & Carbin, 2018; Karnin, 1990; Blalock et al., 2020). It can be broadly classified into two types: *unstructured* and *structured*. Unstructured pruning removes individual weights from the network based on some criteria, such as magnitude or movement, resulting in sparse weight matrices that can be stored and processed more efficiently. Structured pruning, on the other hand, eliminates whole components, such as neurons, channels, or blocks, leading to smaller architectures to reduce end-to-end inference latency. While unstructured pruning has been extensively studied and applied to LLMs (Wang et al., 2020b; Xu et al., 2021; Zafrir et al., 2021; Li et al., 2022), structured pruning is more challenging and less explored. However, structured pruning is also more desirable in many practical scenarios, such as deploying these models on resource-constrained devices or providing fast and reliable services based on LLMs.

Existing work on structured pruning for LLMs focuses on BERT-like networks (Devlin et al., 2018) that consist of an encoder-decoder or an encoder-only architecture (Li et al., 2020; Xia et al., 2022; Zhang et al., 2022; Yao et al., 2021). These models are mainly used for natural language understanding (NLU) tasks, such as question answering, sentiment analysis, or natural language inference. Among the various methods, Block Movement Pruning (Lagunas et al., 2021) is a recent and popular technique that removes weight blocks based on movement. However, there is a lack of systematic research on structured pruning for decoder-only architectures such as GPT-2 Radford et al. (2019), GPT-3 Brown et al. (2020), or GPT-Neo Black et al. (2021), which are mainly used for natural language generation (NLG) tasks, such as text summarization, machine translation, or text completion. While there are some works that apply unstructured pruning (Li et al., 2022) or many kinds of orthogonal compression techniques to decoder-only LLMs (Wang et al., 2020a; Li et al., 2021; Edalati et al., 2022; Tao et al., 2022; Xu & Hu, 2022; Chen et al., 2021), there is no comprehensive evaluation of traditional structured pruning for these models on NLG tasks.

In this work, we compress decoder-only auto-regressive language models. Due to the lack of prior literature towards the same goal, we evaluate the performance of several general-domain pruning methods on NLG tasks, including magnitude and movement pruning. However, we find these methods can struggle or under-perform compared to naïve baselines, leading to the following question:

*What determines the performance of structured pruning on generative language models?*

We aim to fill this gap by conducting a systematic study of structured *fine-pruning* (pruning while finetuning) methods for decoder-only LLMs on NLG tasks[1], and further proposing a novel method that combines the strengths of different existing methods. Our main contributions are:

- To our knowledge, we perform the **first systematic evaluation** of several structured pruning methods to decoder-only LLMs on NLG tasks. We find that they only achieve marginal improvements over *randomly* selecting neurons in finetuning. We explain their limitations under our proposed analysis framework, and characterize their advantages and disadvantages via the metrics we evaluated.

- We propose an empirical analysis framework for structured pruning that relies on two fundamental measures of redundancy: **sensitivity** and **uniqueness**. Sensitivity reflects how much the removal of a network component affects the output of the model, while uniqueness reflects how much the information provided by a network component differs from others. Our framework allows us to understand and compare the behavior and performance of different pruning methods.

- To show the impact made possible by our analysis, we propose a proof-of-concept method, **Globally Unique Movement** (GUM), that aims to maximize both sensitivity and uniqueness by pruning network components based on their global movement and local uniqueness scores. GUM outperforms the existing methods on several NLG tasks and achieves competitive compression rates, proving that future methods should preserve both sensitivity and uniqueness. We also conduct ablation studies to validate the effectiveness of our method components and design choices.

## 2 BACKGROUND & METHODOLOGY

There are many general-domain pruning methods. We focus on *fine-pruning*, a relevant technique for language models which performs automated gradual pruning (Zhu & Gupta, 2017) while fine-tuning. We focus on pruning the MLPs of generative models. At inferencing time, generative models can cache attention vector states. Therefore, especially in large models, MLPs account for more time than attention for new tokens. MLPs also seem to store factual knowledge (Petroni et al., 2019; Meng et al., 2022), making their reduction possibly challenging.

**Notation and Background** We shall define some notations for the MLP layers. Let $\sigma(\cdot) : \mathbb{R}^m \mapsto \mathbb{R}^m$ be an element-wise activation function (e.g. GeLU), and let $W_1 \in \mathbb{R}^{m \times d}, W_2 \in \mathbb{R}^{d \times m}$ be two weight matrices and $b \in \mathbb{R}^m$ be the bias vector. For an input token $x \in \mathbb{R}^d$, the MLP layer output of $x$ is expressed as $\text{MLP}(x) = x + W_2 h(x)$ with intermediate output $h(x) = \sigma(W_1 \text{LN}(x))$, where LN represents layer normalization. We use $\odot$ to denote element-wise multiplications. Lastly, we use $\mathcal{L}$ to denote the loss of the task. We study methods of reducing $m$, the intermediate dimension, which is usually set at $m = 4d$.

**Movement Pruning** Movement Pruning (Sanh et al., 2020) is a popular fine-pruning method. In this paper, we focus on the *block* version of movement pruning (Lagunas et al., 2021), and we first introduce the original unstructured method. Let $\mathcal{L}(W)$ be the task loss with weight parameters $W$. For each weight parameter $W_{i,j}$, we compute a accumulated score $S_{i,j}$ at iteration $T$, by the following expression[2]:

$$S_{i,j}^{(T)} = -\eta_S \sum_{t \leq T} W_{i,j}^{(t)} \cdot \frac{\partial \mathcal{L}(W^{(t)})}{\partial W_{i,j}} \qquad (1)$$

---

[1]All code publicly available at **(removed for peer-review)**.
[2]Gradients are calculated straight-through to the mask scores, otherwise it is undefined (Bengio et al., 2013).

| | Magnitude | Gradual Random | Hard Mvmt. | Soft Mvmt. | GUM |
|---|---|---|---|---|---|
| Score $S$ | $L_2$-norms | Random (frozen) | Eq. 1 | Eq. 1 | Eq. 1 |
| Selection | $\text{Top}_v(S)$ | $\text{Top}_v(S)$ | $\text{Top}_v(S)$ | Threshold | $\text{Top}_v(S)$ |
| Structure | Local | Local | Local | Global | Global |
| Regularization | None | None | $R(S)$ | $R(S)$ | $R(S) + R_{\text{sim}}(S)$ (Eq.4) |
| Criteria | Magnitude | Random | Sensitivity | Sensitivity | Sensitivity&Uniqueness |

Table 1: Comparison of pruning methods used. $R(S)$ is defined in Section 2 and $R_{\text{sim}}(S)$ is described in Eq 4.

Afterwards, the scores are used to compute a mask $M$ with entries $M_{i,j} \in \{0, 1\}$. And we apply the mask by $W' = M \odot W$, $b' = M \odot b$, and $U' = U \odot M$ to remove the masked weights.

There are two ways to compute the mask $M$: $M = \text{Top}_v(S)$ for **hard movement** and $M = \mathbb{1}_{\text{sigmoid}(S) > \tau}$ for **soft movement**. $\tau$ and $v$ are both hyperparameters, and $\text{Top}_v(S)$ is defined as:

$$\text{Top}_v(S)_i = \begin{cases} 1, & S_i \text{ in top } v\%, \\ 0, & \text{otherwise.} \end{cases} \tag{2}$$

Additionally, mask scores are regularized via a regularization term with multiplier $\lambda_{\text{mvp}}$ of the form $R(S) = \lambda_{\text{mvp}} \sum_{i,j} \text{sigmoid}(S_{i,j})$. Hard movement prunes all layers by the same amount. Soft movement, however, allows for adaptive sparsity for different layers, which is known to be crucial for high-sparsity regimes (He et al., 2018b; Mallya et al., 2018), and is seen to be the superior method for NLU tasks in Sanh et al. (2020). Block pruning expands on this method to operate on groups of weights by combining mask scores per block, allowing for structured pruning (Lagunas et al., 2021).

**Magnitude Pruning**   We use a mask version of block magnitude pruning (a block extension of group-lasso, like Shen et al. (2021)) as the baseline. For each set $G$ of parameters, we assign $S_G = (\sum_{(i,j) \in G} |W_{i,j}|^2)^{1/2}$ as the mask score and gradually prune groups with the smallest scores.

**Gradual Random pruning**   Random pruning approaches have been explored previously (Yu et al., 2017; Blalock et al., 2020), and in particular gradual, random pruning (Gale et al., 2019) has been found to perform relatively well. We further explore random pruning in conjunction with distillation. Our gradual random method freezes $S$ at random initialization for the duration of finetuning and prunes using $\text{Top}_v(S)$.

**Knowledge Distillation**   In practice, pruning is often paired with knowledge distillation (Hinton et al., 2015) to boost performance. Distillation loss adds KL divergence between a teacher model and smaller student model. When used, we distill from a finetuned version of the model being pruned.

## 3   FINE-PRUNING FOR GENERATIVE LANGUAGE MODELS

We present our framework for understanding the redundancy of pruning methods. In this work, we focus on improving the seminal work of movement pruning proposed in Sanh et al. (2020). However, naïvely applying movement often results in incremental or worse performance compared to random. We dissect our results using a systematic framework and analyze their behaviors and properties.

### 3.1   OBSERVATIONS OF PREVIOUS PRUNING METHODS

**Soft Movement (BERT's Best Method) Struggles for GPT-like Models**   It is shown in Lagunas et al. (2021) that soft movement enjoys better performance over hard movement when block pruning encoder-decoder models. However, we find the method severely struggles when using the original implementation[3] on GPT-like models due to highly sensitive hyperparameters. For instance, the mask regularization parameter $\lambda_{\text{mvp}}$ can either be too large and prune too aggressively, or too little,

---

[3]Soft movement is the best to our knowledge at time of writing. Code is available at `https://github.com/huggingface/nn_pruning`. In this code, mask scores are added directly to the optimizer and are affected by optimizer algorithm or other hyperparameters. We use this code for a fair comparison between architectures, but manually updating the mask according to the definition might help (Zhang et al., 2022).

| Model | Method | 50% / + Distil | 25% / + Distil | 10% / + Distil |
|---|---|---|---|---|
| | Soft | 63.81 / 65.25 | 63.23 / 65.04 | 62.996* / 64.95* |
| | $L_2$-Magnitude | 62.71 / 65.32 | 61.35 / 64.58 | 61.10 / 63.90 |
| GPT-Neo-125m | Gradual Random | 64.29 / 65.74 | 63.06 / 65.10 | 62.27 / 64.63 |
| Finetuned: **65.92** | Hard/Top$_v$ | 65.33 / 65.94 | 64.88 / 65.79 | 64.23 / 65.17 |
| | GUM | 63.88 / **66.23** | 64.36 / **66.18** | 63.81 / **65.65** |

Table 2: **GPT-Neo-125m:** Performance in $\text{Acc}_{lf}$ on the validation set for decreasing amount leftover on WikiSQL. GUM outperforms compared to other methods, soft movement struggles to match other methods, and gradual random nearly performs as well as Top$_v$. * indicates having 1-3% excess leftover neurons unpruned.

| Model | Method | 50% / + Distil | 25% / + Distil | 10% / + Distil |
|---|---|---|---|---|
| | Soft | 68.27 / 69.319 | 67.74 / 69.314 | 67.25* / 69.11* |
| | $L_2$-Magnitude | 68.33 / 69.62 | 66.92 / 68.96 | 66.02 / 68.43 |
| GPT-2-sm | Gradual Random | 69.07 / 69.61 | 67.78 / 69.35 | 66.77 / 69.00 |
| Finetuned: **70.32** | Hard/Top$_v$ | 69.62 / 69.93 | 69.10 / 69.33 | 68.30 / 69.26 |
| | GUM | 68.62 / **70.38** | 68.82 / **69.63** | 68.07 / **69.46** |

Table 3: **GPT-2-sm:** Performance in $\text{Acc}_{lf}$ on the validation set for decreasing amount leftover on WikiSQL. * indicates having 1-3% excess leftover neurons unpruned.

resulting in under-pruning as shown below. Even after grid searching $\lambda_{\text{mvp}}$ we still find subpar performance, and given the extremely high runtimes for this method as listed in Appendix A, we find this method impractical to use.

**Random Pruning Works Surprisingly Well**  One might expect movement to easily beat random pruning, however, we find their performances to only slightly differ or sometimes match, especially under distillation. Other works have noted random pruning's effectiveness (Gale et al., 2019), but we find the difference in generative tasks to be particularly slim. As shown in Tables 2 and 4, random pruning performs very close to both hard and soft movement pruning over WikiSQL and Wikitext datasets. Moreover, when combined with distillation, the gaps are largely closed between random pruning and other methods, which is also another intriguing observation itself, as we discuss below.

**Distillation Closes the Gaps Between Different Methods**  As shown in Table 2 and Table 3, methods with very different performances would perform rather similar if distilled from a non-pruned, finetuned model. Indeed, both WikiSQL and Wikitext experiments in Table 4 and Table 5 showed that when the network has fewer left-over neurons (e.g., 10% or 25%), the difference of accuracies or perplexities often fall below half of the difference without distillation. This observation remains consistent across models of different sizes, architectures, and tasks. Results for GPT-neo with 1.3-billion parameters in Table 6 shows that pruning a larger model can still benefit from distillation. Knowledge distillation often boosts the performance of weaker methods even more, which might suggest the differences between methods are largely due to the inability to learn more diverse features set in fine-pruning, as suggested by the work of Allen-Zhu & Li (2020).

## 3.2 TWO TYPES OF REDUNDANCY MEASURES: SENSITIVITY AND UNIQUENESS

In order to understand why these pruning methods display such behaviors, we devise a framework to characterize the leftover neurons of pruned network based on two criteria: sensitivity and uniqueness[4]. Sensitivity captures how much a neuron contributes to the task objective $\mathcal{L}$, while uniqueness captures how much information it provides that is not already captured by other neurons. We formalize these notions of redundancy as follows:

---

[4]These are two known concepts in literature, but have not been both combined into one pruning method.

**Definition 3.1 (Redundancy Criteria)** *Given a set of neurons $\{h_i(\cdot)\}_{i\in[m]}$ and input $X$, we call one neuron $h_i$ **redundant** if it meets at least one of the following two conditions:*

1. ***Sensitivity/Saliency**: the neuron is **not salient** if its outputs are either negligible or has small gradient when optimizing for the downstream task, mathematically described as*

$$\mathbb{E}\Big[|h_i(X) \cdot \frac{\partial \mathcal{L}}{\partial h_i(X)}|\Big] \approx 0$$

2. ***Uniqueness**: the neuron is **not unique** if its outputs could be reconstructed entirely with a linear combination of the outputs from other neurons, mathematically described as*

$$h_i(X) \in \mathrm{span}(\{h_j(X)\}_{j\neq i}), \quad over\ all\ inputs\ X$$

Intuitively, a sensitive neuron has outputs that greatly contribute to the final output, while a unique neuron has outputs which are different from that of others. These metrics are independent from one another, so a neuron could be highly salient but replaceable by other neurons, or it could be highly unique but ultimately contribute little to the network. Consider a toy example where two neurons $h_i$ and $h_j$ have the same non-zero weights and large gradient. Neuron $h_i$ could easily be removed by doubling the outputs of $h_j$, so they are not unique, but both are highly salient.

**The General Trade-off Between Saliency and Uniqueness**   Equipped with Definition 3.1, we find the following important trends. Figure 1 and Figure 2 show that without distillation, different methods often have a preference of focus on one of the redundancy measures. We now comment on trends across all experimental result tables (Table 2, 3, 4, 5, 6, 7, 8). The best performing methods strike a balance between both measures, establishing a strong correlative link between these metrics and final pruning performance. Under distillation, however, sensitivity seemingly concentrates across methods. Regardless of method, as more pruning occurs, sensitivity decreases and uniqueness increases in general. For individual methods, we can further dive deeper as below:

- *Magnitude pruning* universally scores worst on both metrics, explaining its poorer performance in all experiments. However, with distillation, gaps of sensitivity between methods noticeably decreases, which partially describes why distillation improves it significantly.

- *Random pruning* obtains similar distillation sensitivity and uniqueness, though slightly lower, to hard movement, lending credence to its overall high performance. However, sensitivity is markedly lower without distillation as is reflected in all figures. *This is proof that hard movement does not target uniqueness*, given random pruning does not target uniqueness.

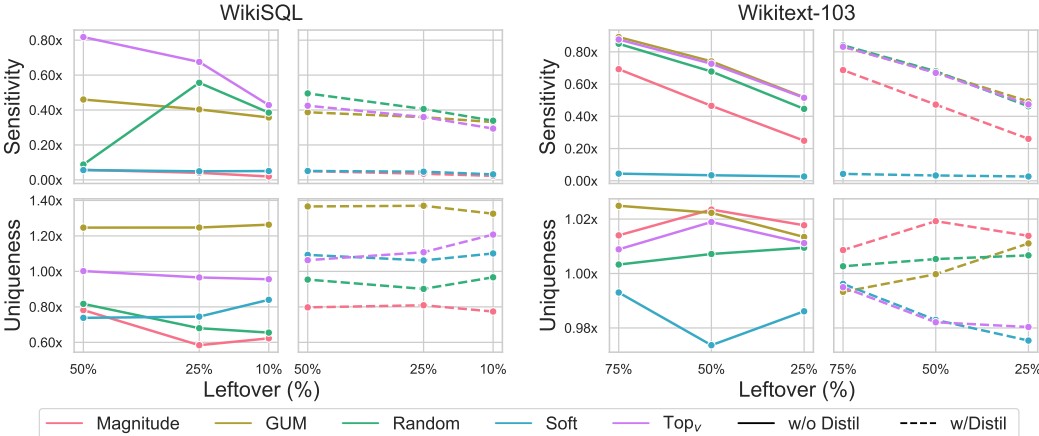

Figure 1: Sensitivity and Uniqueness measured on the training set for GPT-Neo-125m. The vertical axis is defined as the ratio of the corresponding metric between the pruned model and a baseline model (which is non-pruned and fully fine-tuned) with a maximum of 1x. We are able to use these graphs to analyze and compare the performance of different pruning methods. Details of measurements are given in Appendix E.

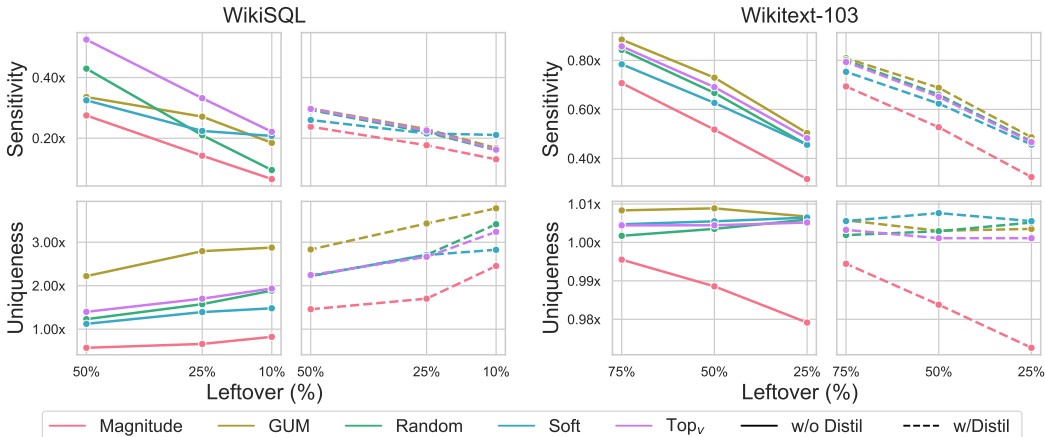

Figure 2: Sensitivity and Uniqueness measured on the training set for GPT-2-sm. The vertical axis is defined as the ratio of the corresponding metric between the pruned model and a baseline model (which is non-pruned and fully fine-tuned) with a maximum of 1x. Details of measurements are given in Appendix E.

- *Soft movement pruning* also usually scores poorly on both metrics, and sometimes abysmally as in its sensitivity in Figure 1, helping describe its overall poor performance.
- *Hard movement pruning* consistently obtains the highest sensitivity with not-far-behind uniqueness across different datasets and architectures. This correlates with the high performance when distillation is not used. However, when combined with distillation, the gaps of sensitivity between methods converge, and the advantage of hard movement fades.
- *GUM*, our proof-of-concept method, nearly always obtains best uniqueness while maintaining decent sensitivity, further improved using distillation, explaining its superiority across various tasks. However, GUM has a larger performance increase for GPT-Neo-125m than for GPT-2-sm on WikiSQL; this is explained in Figure 2 as pruned GPT-2 already has high baseline uniqueness for WikiSQL ($\sim$2x) so further increase incurs diminishing returns.

Given the training/validation split and general noise in the datasets, there are some outlier points, for instance, GUM's surprisingly poor distilled uniqueness for Wikitext in Figure 1. We observe higher absolute uniqueness on Wikitext in general (around 95% of neurons are unique per cosine similarity), meaning uniqueness varies over datasets and improving uniqueness is difficult.

## 4   GLOBALLY UNIQUE MOVEMENT

After observing the lack of uniqueness amongst leftover neurons, we set out to improve the performance of hard movement. We introduce two techniques which together comprise Globally Unique Movement (GUM). In essence, we encourage a score-weighted uniqueness term by multiplying the score regularizer and the cosine similarity together, to obtain a balance of uniqueness and sensitivity.

**Tackling Non-Unique Redundancy**   Regularizing or pruning via similarity is a well-explored topic (Ayinde et al., 2019; Zhu et al., 2018; Srinivas & Babu, 2015; Santacroce et al., 2020) and existing techniques would increase uniqueness. However, we integrate more cleanly with movement to insulate weights from regularization, with a small increase in training time as listed in Appendix A.

Our approach regularizes mask scores based on cosine similarity [5]. Cosine similarity between the outputs of any two neurons given input $X$ (for example, a batch of tokens) is defined simply as

$$\text{sim}(h_i(X), h_j(X)) = \frac{h_i(X)^\top h_j(X)}{\|h_i(X)\|_2 * \|h_j(X)\|_2} \tag{3}$$

---

[5]Solving for linear combinations of neurons during training is prohibitively expensive, so we consider cosine similarity as a "first-order" proxy.

---

**Algorithm 1** Running Cosine Similarity Update

---

**Require:** a set of neurons $h_i(\cdot)$ for $i \in [m]$, inputs from a set $\mathcal{Z}$ (usually intermediate outputs of attention layers), an update multiplier $\lambda_{\text{sim}}$;

1: Initialize running similarity $\mathbf{sim}^{(0)}(h_i, h_j) = 0$ for $i, j \in [m]$, running inner products $C_{i,j}^{(0)} = 0$, and running output vector norms $Q_i^{(0)} = 0$ for $i \in [m]$;

2: **while** still training **do**

3:     Sample a input $X \in \mathcal{Z}$, compute the output vector neuron $h_j(X)$ for $j \in [m]$[7].

4:     Update $C_{i,j}^{(t+1)} \leftarrow (1 - \lambda_{\text{sim}})C_{i,j}^{(t)} + \lambda_{\text{sim}} \cdot h_i(X)^\top h_j(X), \quad \forall i, j \in [m]$

5:     Update $Q_i^{(t+1)} \leftarrow (1 - \lambda_{\text{sim}})Q_i^{(t)} + \lambda_{\text{sim}} \cdot \|h_i(X)\|_2^2, \quad \forall i \in [m]$

6:     Update similarity by:

$$\mathbf{sim}^{(t+1)}(h_i, h_j) \leftarrow C_{i,j}^{(t+1)} / \sqrt{Q_i^{(t+1)} Q_j^{(t+1)}}, \quad \forall i, j \in [m]$$

7: **end while**

---

However, calculating similarity with only intra-batch estimates is noisy and unreliable, so we introduce a running version of its estimates in Algorithm 1 to obtain cosine similarity $\mathbf{sim}^{(t)}(h_i, h_j)$ between neurons $h_i(\cdot)$ and $h_j(\cdot)$. Now, we build on the regularization term $R_{\text{sim}}(S)$ of movement pruning to define a new regularization: Let $N_{\text{left}}$ be the number of leftover neurons, for each group $j \in [m]$ and its corresponding score $S_j$, we define a term $U_j = \frac{1}{N_{\text{left}}} \sum_{i \in [m], i \neq j} \mathbf{sim}(h_j, h_i)$, and then we multiply $U_j$ to the original terms in $R(S)$ to obtain

$$R_{\text{sim}}(S) = \lambda_{\text{mvp}} \sum_j U_j \cdot \text{sigmoid}(S_j) \tag{4}$$

**Global Top$_v$ for Soft-Like Movement**    Hard movement removes the same amount of weights per layer independently. Global Top$_v$ instead uses Top$_v$ function on the set of all mask scores in the network jointly. Global Top$_v$ was originally explored for movement (Sanh et al., 2020) and was found to perform similarly. We find when used in conjunction with uniqueness regularization, Global outperforms Local. Global Top$_v$ intuitively allows for more flexibility when pruning. When pruning locally, it is necessary to choose the least common pruning percent - if one layer requires 50% neurons before severe degradation, all layers must keep 50%. Global comparison removes this loophole in a similar manner to soft movement [6].

## 5    RESULTS

In this section, we present results on three different kinds of generative language modeling tasks: language modeling with Wikitext-103 Merity et al. (2016), text-to-text generation and natural language understanding with SAMsum Gliwa et al. (2019), and exact match measurable text-to-code generation with WikiSQL Zhong et al. (2017). Details and hyperparameters are listed in Appendix F. When distilling, the teacher model used is the finetuned version of the model. To ensure trends hold when scaling up, we present one experiment with GPT-Neo-1.3b in section 5. For all pruning amounts, we will present in terms of final percentage *leftover* - i.e., 75% of neurons remain after pruning. For soft movement, final prune percentage is shown in parentheses when it differs from desired by a large amount.

In general, GUM is found to outperform Top$_v$ by a margin similar to the difference between Top$_v$ and gradual random pruning, with some exceptions. While small, we argue this gap shows the effectiveness of preserving neuron uniqueness alongside saliency.

**Wikitext-103**    Results on the Wikitext-103 dataset Merity et al. (2016), one of the most popular datasets for causal language modeling, are shown in Tables 4 and 5. Because performance on Wikitext-103 is in perplexity (PPL), it is a highly consistent and discriminatory dataset to prune on. We are unable to ever fully recover original model performance after pruning, suggesting that any compression increases uncertainty. Distillation generally hurts performance across all methods.

---

[6]Appendix D shows an example pruning distribution for one pruned network.

| Model | Method | 75% / + Distil | 50% / + Distil | 25% / + Distil |
|---|---|---|---|---|
| | Soft | 17.814 / 17.651 | 19.470 / 19.053 | 21.169 / 20.4678* |
| | $L_2$-Magnitude | 18.524 / 18.048 | 20.834 / 20.041 | 23.692 / 22.604 |
| GPT-Neo-125m | Gradual Random | 17.307 / 17.144 | 18.900 / 18.410 | 21.458 / 20.546 |
| Finetuned: **16.138** | Hard/Top$_v$ | 16.974 / 17.142 | 18.253 / 18.369 | 20.495 / 20.194 |
| | GUM | **16.822** / 17.158 | **17.881** / 18.314 | 20.059 / **19.833** |

Table 4: **GPT-Neo-125m:** Performance in perplexity (PPL) on the validation set for decreasing amount leftover on Wikitext-103. * indicates having 1-3% excess leftover neurons unpruned.

| Model | Method | 75% / + Distil | 50% / + Distil | 25% / + Distil |
|---|---|---|---|---|
| | Soft | 16.754 / 16.950 | 18.261 / 18.051 | 19.948 / **19.473**∗ |
| | $L_2$-Magnitude | 17.399 / 17.414 | 19.595 / 19.178 | 22.593 / 21.667 |
| GPT-2-sm | Gradual Random | 16.574 / 16.823 | 17.974 / 17.862 | 20.444 / 19.798 |
| Finetuned: **15.571** | Hard/Top$_v$ | 16.363 / 16.730 | 17.611 / 17.742 | 20.016 / **19.663** |
| | GUM | **16.242** / 16.680 | **17.444** / 17.692 | 19.877 / 19.681 |

Table 5: **GPT-2-sm:** Performance in perplexity (PPL) on the validation set for decreasing amount leftover on Wikitext-103. * indicates having 1-3% excess leftover neurons unpruned.

**WikiSQL**   As opposed to the other datasets, WikiSQL Zhong et al. (2017) contains hard ground-truth labels for comparison via Exact Match (EM). Due to this, our best performance is achieved in WikiSQL, where GUM is able to remove up to 75% of neurons while maintaining performance on GPT-Neo. Results are shown in Tables 2 and 3. We also present results for GPT-Neo-1.3b only on WikiSQL in Table 6. Results for this experiment follow a similar trend to smaller models.

| Model | Method | 50% / + Distil | 25% / + Distil | 10% / + Distil |
|---|---|---|---|---|
| GPT-Neo-1.3B | Gradual Random | 72.18 / **74.76** | 70.38 / 73.83 | 68.56 / 72.77 |
| Finetuned: **74.88** | Hard/Top$_v$ | 73.33 / 74.75 | 72.18 / 74.54 | 71.14 / 73.77 |
| | GUM | 72.88 / 74.70 | 71.80 / **74.62** | 71.35 / **74.157** |

Table 6: **GPT-Neo-1.3B:** Performance in Acc$_{lf}$ for decreasing amount leftover on WikiSQL.

**SAMsum**   Results on SAMsum Gliwa et al. (2019) are presented in Tables 7 and 8. Popular for encoder-decoder models, this dataset entails summarizing short instant message conversations. Larger generative models have been explored for this task (Feng et al., 2021; Zhang et al., 2019), achieving competitive results. We use this dataset to test the natural language understanding and summarization skills of small models under pruning. We note poor relative baseline results relative to encoder-decoder models as expected, however, pruning trends follow that of other datasets and GUM generally outperforms Top$_v$.

## 6   ADDITIONAL RELATED WORKS

**General Domain Pruning**   Neural net pruning has been proposed years before the explosion of deep learning research (Janowsky, 1989; Mozer & Smolensky, 1988; Karnin, 1990), and are summarized in an outdated survey Reed (1993). Previous works have explored many approaches of pruning neural nets (Wen et al., 2016; Han et al., 2015b;a; Li et al., 2016). Recently, the *lottery ticket hypothesis* Frankle & Carbin (2018) proposed a new direction to prune at initialization instead. However, there is also a massive divergence of methods or claims that it is not worthwhile Liu et al. (2018); Blalock et al. (2020). Regardless, many strong techniques exist in modern incarnations across all kinds of architectures (Yang et al., 2016; Luo et al., 2017; He et al., 2018a).

| Method | Leftover % | Rouge_1 | Rouge_2 | Rouge_L | Rouge_LSUM |
|---|---|---|---|---|---|
| No Prune | | 38.68 | 14.74 | 31.73 | 31.76 |
| Gradual Random | 50% / + Distil | 35.54 / 36.82 | 12.71 / 13.40 | 29.11 / 30.28 | 29.04 / 30.27 |
| | 25% / + Distil | 33.11 / 35.71 | 11.01 / 13.13 | 27.39 / 29.50 | 27.37 / 29.48 |
| | 10% / + Distil | 31.83 /34.60 | 10.02 / 11.72 | 26.49 / 28.40 | 26.46 / 28.34 |
| Hard/Top$_v$ | 50% / + Distil | 37.68 / 36.94 | 14.17 / 13.72 | 31.12 / 30.64 | 31.09 / 30.62 |
| | 25% / + Distil | 36.38 / 37.34 | 13.00 / **14.24** | 29.96 / **31.17** | 29.95 / **31.15** |
| | 10% / + Distil | 33.07 / 36.12 | 10.95 / 12.62 | 27.70 / 29.70 | 27.68 / 29.67 |
| GUM | 50% / + Distil | 37.22 / **38.45** | 13.79/ **14.27** | 30.72 / **31.35** | 30.72 / **31.36** |
| | 25% / + Distil | 36.18 / **37.57** | 13.18 / 13.71 | 29.99 / 30.91 | 30.00/ 30.93 |
| | 10% / + Distil | 34.72 / **36.52** | 11.88 / **13.40** | 28.82 / **29.97** | 28.79 / **29.96** |

Table 7: **GPT-Neo-125m:** Validation results on SAMsum. Higher is better for all metrics. In general, more pruning hurts performance, and GUM outperforms Top$_v$.

| Method | Leftover % | Rouge_1 | Rouge_2 | Rouge_L | Rouge_LSUM |
|---|---|---|---|---|---|
| No Prune | | 40.83 | 16.85 | 33.72 | 33.70 |
| Gradual Random | 50% / + Distil | 39.29 / 39.69 | 15.25 / 15.74 | 32.16 /32.72 | 32.15 / 32.72 |
| | 25% / + Distil | 38.09 / 39.73 | 14.43 / 15.35 | 31.16 / 32.72 | 31.15 / 32.66 |
| | 10% / + Distil | 36.91 / 38.88 | 13.02 / 14.71 | 30.13 / 31.85 | 30.13 / 31.83 |
| Hard/Top$_v$ | 50% / + Distil | 39.93 / 40.23 | 16.43 / 16.07 | 33.40 / 33.39 | 33.38 / 33.39 |
| | 25% / + Distil | 38.84 / **40.49** | 14.88 / **16.12** | 31.83 / **33.46** | 31.82 / **33.45** |
| | 10% / + Distil | 38.28 / 39.49 | 14.58 / 15.32 | 31.36 / 32.57 | 31.37 / 32.57 |
| GUM | 50% / + Distil | 38.80 / **40.74** | 15.52 / **16.22** | 32.27 / **33.99** | 32.23 / **33.95** |
| | 25% / + Distil | 37.56 / 39.74 | 14.24 / 15.72 | 30.94 / 32.93 | 30.91 / 32.93 |
| | 10% / + Distil | 39.12 / **39.80** | 15.01 / **15.79** | 32.20 / **32.96** | 32.21 / **32.95** |

Table 8: **GPT-2-sm:** Validation results on SAMsum. Higher is better for all metrics. In general, more pruning hurts performance, and GUM outperforms Top$_v$.

**Compressing Language Models** Compressing LLMs in particular has spawned a unique kind of pruning. Given that LLMs first undergo pre-training on massive amounts of data, works such as Xu et al. (2021); Zafrir et al. (2021); Li et al. (2022) find ways to prune and finetune these models on downstream data. Building on automated gradual pruning (Zhu & Gupta, 2017) and learned threshold pruning (Azarian et al., 2020), movement pruning (Sanh et al., 2020) and further block movement (Lagunas et al., 2021) have become highly popular methods for *fine-pruning*, combining finetuning with pruning for overall best performance. Since then, many works have attempted to improve on movement (Yao et al., 2021; Zhang et al., 2022; Xia et al., 2022; Kwon et al., 2022). As previously mentioned, however, we are unable to find any comparable works systematically exploring structured pruning for decoder-only models.

## 7 CONCLUSION & FUTURE WORK

In this paper, we have performed an evaluation of structured pruning on generative language models, finding existing methods to improve over random pruning less than expected. In addition, we have proposed a framework for analyzing pruning methods in terms of uniqueness and saliency, two important criteria for preserving model quality and diversity. We have presented a novel method based on these metrics, GUM, for structured pruning of generative models, based on uniqueness regularization and global Top$_v$ pruning. Our method can be applied to the MLP layers of various generative models, but there are still many open questions and challenges for pruning other components, such as attention heads or MoE modules. We also acknowledge the limitations of our method, which can reduce saliency and performance, suggesting possible directions for improving uniqueness pruning. Our work is an initial step towards understanding and improving structured pruning of generative models.

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

## A    COMPUTATIONAL RUNTIME COMPARISON

The training runtime of all pruning methods are compared in Tables 9, 10, and 11. For all experiments, Soft movement has a greatly increased runtime compared to other pruning methods. This is due to the over-pruning problem previously described: if soft movement prunes too many neurons, it must default to hard movement as a backup. To reach a specific pruning percentage, this must occur, resulting in significantly more computation.

GUM is certainly slower than hard movement, however, we find the difference to be minimal. Compared to no pruning, all pruning methods add a significant amount of time to training, especially when also combined with distillation. Therefore, the additional runtime incurred by GUM is small in comparison.

| Model | No Prune | $L_2$ | Random | Hard | GUM | Soft |
|---|---|---|---|---|---|---|
| GPT-Neo-125m | 4.75 | 10.5 | 9.93 | 10.26 | 11.46 | 25.33 |
| GPT-Neo-125m + Distil | - | 13.83 | 13.83 | 13.93 | 15.4 | 31.5 |
| GPT-2-sm | 6.83 | 11.62 | 10.06 | 10.07 | 12.83 | 27.5 |
| GPT-2-sm + Distil | - | 9.28 | 14.22 | 14.55 | 17.08 | 32.67 |
| GPT-Neo-1.3b | 3.92 | - | 11.83 | 12.25 | 12.83 | - |
| GPT-Neo-1.3b + Distil | - | - | 14.5 | 14.8 | 15.56 | - |

Table 9: **WikiSQL Training Runtime in Hours**. GPT-Neo-125m and GPT-2-sm were run on 8xV100 GPUs, while GPT-Neo-1.3b was run on 8xA100 GPUs. All results are averaged over all pruning runs, which are comparable given neuron removal occurs after training.

| Model | No Prune | $L_2$ | Random | Hard | GUM | Soft |
|---|---|---|---|---|---|---|
| GPT-Neo-125m | 5.42 | 8.55 | 6.82 | 7.1 | 8.87 | 18.08 |
| GPT-Neo-125m + Distil | - | 11.9 | 11.9 | 11.8 | 13.45 | 23.33 |
| GPT-2-sm | 4.72 | 6.45 | 6.17 | 6.12 | 9.1 | 17.13 |
| GPT-2-sm + Distil | - | 11.33 | 10.88 | 11.12 | 13.48 | 22.17 |

Table 10: **Wikitext Training Runtime in Hours**. GPT-Neo-125m and GPT-2-sm were run on 8xV100 GPUs. All results are averaged over all pruning runs, which are comparable given neuron removal occurs after training.

| Model | No Prune | Random | Hard | GUM |
|---|---|---|---|---|
| GPT-Neo-125m | 1.72 | 2.22 | 2.37 | 2.71 |
| GPT-Neo-125m + Distil | - | 4.45 | 5.13 | 5.33 |
| GPT-2-sm | 1.55 | 2.41 | 2.27 | 2.79 |
| GPT-2-sm + Distil | - | 3.96 | 3.93 | 4.17 |

Table 11: **SAMsum Training Runtime in Hours**. GPT-Neo-125m and GPT-2-sm were run on 8xV100 GPUs. All results are averaged over all pruning runs, which are comparable given neuron removal occurs after training.

## B   E2E NLG CHALLENGE RESULTS

We also tested pruning on the E2E NLG Challenge Dušek et al. (2020) in Tables 12 and 13. A highly popular dataset, performance in this domain is measured by a variety of metrics that all measure the quality of the output indirectly as opposed to the direct measurement versus ground truth in other experiments.

After many rounds of hyperparameter optimization, results for GPT-2-sm loosely follow previously seen trends, however, GPT-Neo-125m results are highly inconsistent. For this model removing even 90% of neurons can result in best performance, and increasing pruning does not monotonically affect performance, two highly concerning phenomenon unseen in other datasets.

We speculate these inconsistencies could be due to many reasons; such as the open-endedness of the problem domain, or too little training data. Measuring the performance of generative models is a non-trivial task, especially when model outputs are highly similar as is the case when pruning.

For these reasons, we do not further experiment on this dataset and leave it out of our main analysis. However, we include this experiment to show that in some cases, pruning results can be inconsistent for language models. Further exploration is required into this area.

| Method & Leftover % | BLEU | NIST | METEOR | ROUGE_L | CIDEr |
|---|---|---|---|---|---|
| No Prune | 68.22 | 8.6315 | 0.4479 | 0.7103 | 2.3141 |
| $\text{Top}_v$ 75% | 68.26 | 8.6660 | 0.4486 | 0.7108 | 2.3120 |
| $\text{Top}_v$ 25% | 68.80 | 8.6780 | 0.4487 | 0.7126 | 2.3238 |
| $\text{Top}_v$ 10% | 68.15 | 8.6377 | 0.4472 | 0.7144 | 2.2955 |
| GUM 75% | 68.07 | 8.5698 | 0.4458 | 0.7090 | 2.2835 |
| GUM 25% | 68.22 | 8.6597 | 0.4495 | 0.7093 | 2.3230 |
| GUM 10% | 68.26 | 8.6684 | 0.4490 | 0.7109 | 2.3012 |

Table 12: **GPT-Neo-125m:**   Testing results on the E2E NLG Challenge. Higher is better for all metrics. Even at 10% leftover, performance is similar to the baseline, for both $\text{Top}_v$ and GUM.

| Method & Leftover % | BLEU | NIST | METEOR | ROUGE_L | CIDEr |
|---|---|---|---|---|---|
| No Prune | 68.05 | 8.6547 | 0.4623 | 0.7143 | 2.4500 |
| $\text{Top}_v$ 75% | 66.79 | 8.5064 | 0.4590 | 0.7119 | 2.4308 |
| $\text{Top}_v$ 25% | 66.85 | 8.5115 | 0.4587 | 0.7098 | 2.4322 |
| $\text{Top}_v$ 10% | 66.63 | 8.4783 | 0.4588 | 0.7091 | 2.4368 |
| GUM 75% | 68.02 | 8.6552 | 0.4619 | 0.7134 | 2.4502 |
| GUM 25% | 66.58 | 8.4853 | 0.4588 | 0.7098 | 2.4286 |
| GUM 10% | 67.06 | 8.5237 | 0.4601 | 0.7122 | 2.4417 |

Table 13: **GPT-2-sm:** Testing results on the E2E NLG Challenge. Higher is better for all metrics. In general, more pruning results in worse performance, and GUM outperforms $\text{Top}_v$.

## C  UNIQUENESS REGULARIZATION PER LAYER

The goal of uniqueness regularization is to punish similarity between neurons, especially those with high similarity. Figure 3 shows that GUM is generally successful in removing neurons with high similarity across all layers.

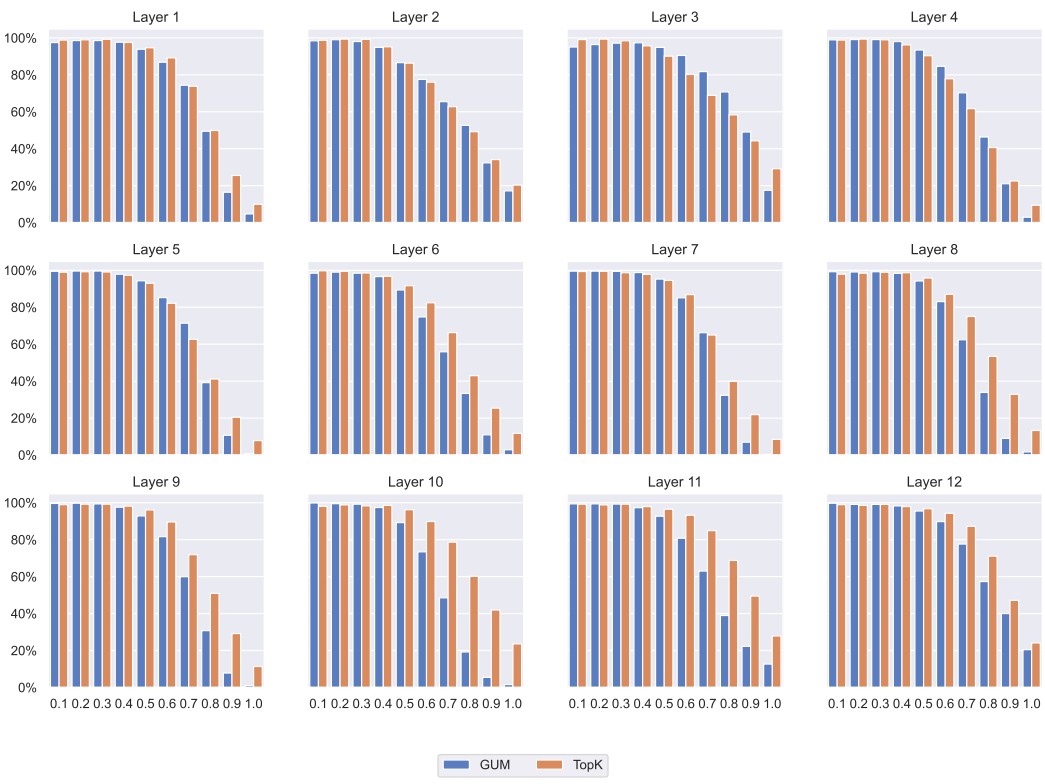

Figure 3: For each layer, this graph shows the percentage of neurons with at least one similarity per range. Similarity is defined as the absolute value of cosine similarity over the entire validation dataset, increasing from 0 to 1. $\text{Top}_v$ and GUM are compared, training on WikiSQL with GPT-Neo-125m. Total leftover neurons is exactly 25% of all neurons.

## D  GLOBAL $\text{TOP}_v$ REMOVAL PER LAYER

A natural question arises with Global $\text{Top}_v$ pruning: what is the final prune percentage per-layer? Figure 4 shows the prune percentages per layer for one sample training run. From this test, later layers are clearly prioritized, with a large emphasis on the last layer. While the exact layers pruned more or less will vary with noise, model, and dataset, we observe this trend to generally hold true.

## E  UNIQUENESS AND SENSITIVITY GRAPHING

To measure sensitivity for a model, we measure the global sum of sensitivity for all neurons $\{h_i(\cdot)\}_{i \in [m]}$ in feedforward layers $\{l \in L\}$ on the training dataset via (taking the absolute value to account for sign):

$$\Sigma_l \Sigma_i |h_i(X) \cdot \frac{\partial \mathcal{L}}{\partial h_i(X)}|$$

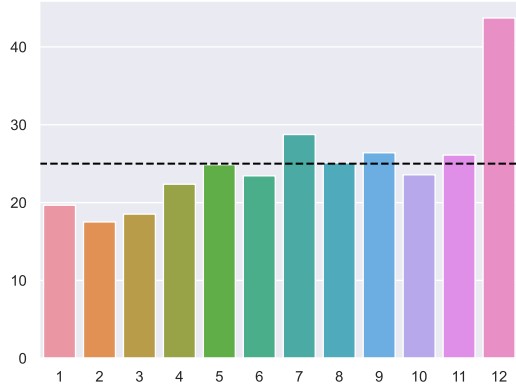

Figure 4: The percentage leftover for layers 1-12 after Global Top$_v$ pruning, using GUM training on WikiSQL with GPT-Neo-125m. Total leftover neurons is exactly 25% of all neurons.

To measure uniqueness for a model, we measure the cosine similarity of each neuron with each other neuron as in equation 3 over the entire dataset, measuring each pair only once. This can be measured using the running cosine similarity without a decay multiplier. Then, we measure the percentage of neurons with at least 1 similarity above 0.8 to another neuron (i.e. these neurons nearly match each other) across the entire network.

Both sensitivity and uniqueness are not useful on their own, but are useful relative to other models. We therefore divide both metrics by the value obtained for the fully finetuned version of the model.

Overall, uniqueness values for Wikitext-103 and for WikiSQL were quite different. Neurons on Wikitext-103 seem to already be highly unique, with some specific layers containing a large amount of redundancy, while WikiSQL has redundancy throughout the entire network.

Figures 5 and 6 show non-re-scaled results for the previous graphs. These graphs show uniqueness as a percent of all neurons (i.e. 50% of neurons are unique) and sensitivity as a raw value.

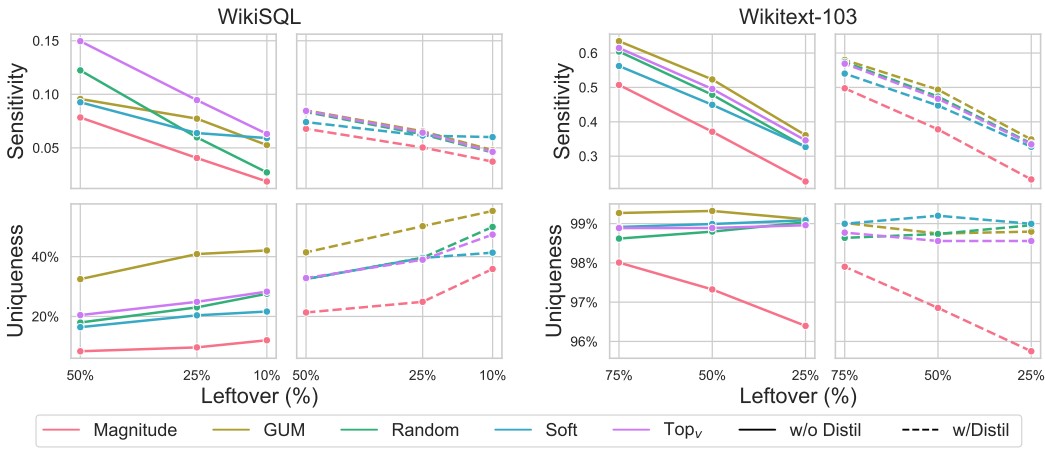

Figure 5: Sensitivity and Uniqueness measured without re-scaling, for GPT-2-sm.

## F  TRAINING HYPERPARAMETERS

All training hyperparameters are provided. Only one random seed was attempted per training run. We note that it is possible different pruning amounts require different hyperparameters and therefore performance could be better, however, searching for each possible combination would be far too

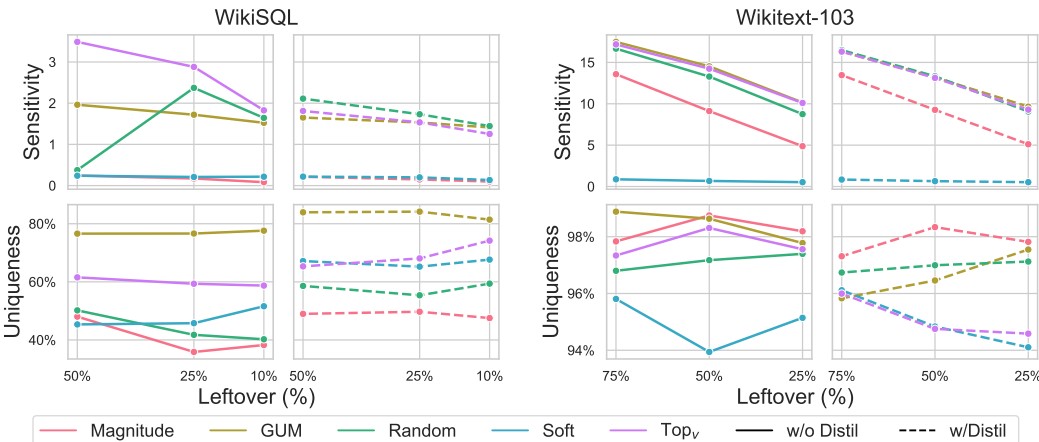

Figure 6: Sensitivity and Uniqueness measured without re-scaling, for GPT-Neo-125m.

expensive. Therefore, the same hyperparameters are used for different pruning percentages. Only the best combination for each model is listed, but each model had its hyperparameters tuned manually. We explored hyperparameters by starting with known vaules from literature, then performing grid searches on relevant pruning vaules (i.e. distillation temperature, or $\lambda_{gum}$).

L1 regularization was used on the mask scores for all models. Learning weight decayed linearly to 0 for all models. LR means Learning Rate, WD means Weight Decay, and LS means Label Smoothing. When distillation was used, the teacher was trained using the same hyperparameters (sans pruning arguments).

| Adam $\beta_1$ | Adam $\beta_2$ | Adam $\epsilon$ | Batch | LR Warmup Percent | GUM $\lambda_c$ |
|---|---|---|---|---|---|
| 0.9 | 0.999 | 1e-4 | 8 | 10% | 0.99 |

Table 14: Shared hyperparameters for all runs.

**WikiSQL** A max token length of 512 was used. Strings were not converted to lowercase. Special tokens were added by the tokenizer.

GPT-Neo-125m used a mask LR of 1e-2 for Hard Movement/GUM and 1e1 for Soft Movement. GPT-2-sm used a mask LR of 1e-2 for Hard Movement/GUM and 1e1 for Soft Movement. Soft movement required a large mask learning rate as it would otherwise not converge - other combinations of more regularization were attempted.

| Model | LR | WD | Epochs | $\lambda_{mvp}$ | $\lambda_{gum}$ | LS | Distil $\alpha$ | Distil Temp. |
|---|---|---|---|---|---|---|---|---|
| GPT-Neo-125m | 5e-4 | 0.05 | 10 | 2 | 1e2 | 0.05 | 0.5 | 2 |
| GPT-2-sm | 3e-4 | 0.1 | 11 | 2 | 1e1 | 0.05 | 0.9 | 1 |
| GPT-Neo-1.3b | 2e-4 | 0.05 | 6 | 2 | 1e1 | 0 | 0.5 | 1 |

Table 15: WikiSQL hyperparameters.

**Wikitext-103** A max token length of 1024 was used. Strings were not converted to lowercase. Special tokens were added by the tokenizer.

GPT-Neo-125m used a mask LR of 1e-2 for Hard Movement/GUM and 1e1 for Soft Movement. GPT-2-sm used a mask LR of 1e-2 for Hard Movement/GUM and 1e1 for Soft Movement. Soft

movement required a large mask learning rate as it would otherwise not converge - other combinations of more regularization were attempted.

For Wikitext-103 specifically, Soft Movement also required an extremely large $\lambda_{mvp}$. This causes a large increase in pruning at the beginning of training, which could explain the overall poor performance. Without this large value, however, Soft Movement would not converge to the desired pruning amount.

| Model | LR | WD | Epochs | $\lambda_{mvp}$ | $\lambda_{gum}$ | LS | Distil $\alpha$ | Distil Temp. |
|---|---|---|---|---|---|---|---|---|
| GPT-Neo-125m | 5e-4 | 0.05 | 3 | 2, 1e2 (Soft) | 1e1 | 0.05 | 0.9 | 1 |
| GPT-2-sm | 3e-4 | 0.1 | 3 | 2, 1e2 (Soft) | 1e1 | 0.05 | 0.9 | 1 |

Table 16: Wikitext-103 hyperparameters.

**SAMsum** A max token length of 1024 was used. Strings were not converted to lowercase, and whitespace was not stripped. More than 3 or 4 epochs starts to result in overtraining for both models, so both were limited to not overtrain.

For GPT-Neo-125m, both methods used a mask LR of 1e-2. For GPT-2-sm, GUM required a mask LR of 1e-3, while TopK used a mask LR of 1e-2. However, for both models $\lambda_{gum}$ must be 1e1 for no distillation, and 1e2 for distillation, as too high a $\lambda_{gum}$ results in poor non-distil performance. Soft movement was not attempted on this dataset.

| Model | LR | WD | Epochs | $\lambda_{mvp}$ | $\lambda_{gum}$ | LS | Distil $\alpha$ | Distil Temp. |
|---|---|---|---|---|---|---|---|---|
| GPT-Neo-125m | 5e-4 | 0.01 | 4 | 2 | 1e1/1e2 | 0.01 | 0.5 | 1 |
| GPT-2-sm | 3e-4 | 0.05 | 6 | 2 | 1e1 | 0.01 | 0.9 | 1 |

Table 17: SAMsum hyperparameters.

**E2E** A max token length of 512 was used. When testing, BEAM search was used with 10 beams, a length penalty of .9, and no ngram repeat size of 4.

GPT-Neo-125m used a mask LR of 1e-2 for Hard Movement/GUM and 1e1 for Soft Movement. GPT-2-sm used a mask LR of 1e-2 for Hard Movement/GUM and 1e1 for Soft Movement. Soft movement required a large mask learning rate as it would otherwise not converge - other combinations of more regularization were attempted.

| Model | LR | WD | Epochs | $\lambda_{mvp}$ | $\lambda_{gum}$ | LS | Distil $\alpha$ | Distil Temp. |
|---|---|---|---|---|---|---|---|---|
| GPT-Neo-125m | 5e-4 | 0.01 | 6 | 2, 1e1 (Soft) | 1e1 | 0.05 | 0.5 | 2 |
| GPT-2-sm | 3e-4 | 0.1 | 6 | 2, 1e2 (Soft) | 1e1 | 0.05 | 0.9 | 1 |

Table 18: E2E NLG Challenge hyperparameters.

