# OpenReview forum: "What Matters In The Structured Pruning of Generative Language Models?"
_ICLR.cc/2023/Conference — Submitted to ICLR 2023_

### Official Review · Reviewer_25Qh · 2022-10-21

**Confidence:** 4
**Correctness:** 3
**Technical Novelty And Significance:** 3
**Empirical Novelty And Significance:** 3
**Recommendation:** 5

**Clarity, Quality, Novelty And Reproducibility:**

Overall, the paper sets out to tackle an important problem and provides comprehensive empirical results. The proposed method is novel and interesting although it's not well justified whether the minor improvements outweigh the extra complexity introduced. For experiments completeness, the authors should evaluated on other more common tasks/datasets as is often experimented with causal language models.

Some of the experimentation details could use more clarifications:
- The authors could provide more motivation around why pruning is only performed on MLP layers.
- Different from other methods, GUM does not benefit from combining with distillation on language modeling (Table 4 & 5). Any insights on why?
- Could you provide comparison on training speed (GUM vs. other methods)?
- The improvement from GUM are small in most cases. Are they statistically significant? Could you provide results from multiple runs with different seeds and show the variance?

**Strength And Weaknesses:**

Strengths:
- This paper studies an important problem, i.e. pruning LLMs, for increasing the accessibility of LLMs as they keep growing larger and larger. This problem is also under-explored so far where both empirical results and theoretical understandings are lacking.
- This authors carefully analyzed existing methods through neuron redundancy metrics which shed lights on their performance in pruning.
- The proposed method is novel and interesting.
- The writing is clear and easy to follow.

Weaknesses:
- The evaluation tasks and datasets do not have full coverage of how generative LLM are often evaluated. The authors evaluated on three tasks: language modeling (wikitext-103), text-to-code generation (wikiSQL), text-to-text generation (E2E NLG challenge). There are a couple of problems. First, E2E NLG challenge is not very diagnostic as the authors found that "E2E NLG challenge is not very diagnostic " and "speculate these discrepancies are due to the open-endedness of the problem domain and underfitting the data." I would expect the authors to evaluate on a more robust text-to-text task (e.g. summarization) to derive more informative evaluation results. Second, besides language modeling and text-to-code generation, LLMs are often used for language understanding and reasoning tasks, e.g. those evaluated in the GPT-3 paper. Experiments on pruning should be conducted on some of those tasks.
- The proposed approach computes cosine similarity for any two neurons' outputs. Although it's a sound solution, it is computationally expensive. Also, I wonder whether the running cosine similarity has high variance and how it affects the effectiveness of pruning.
- Improvements from GUM are minor especially on language modeling.
- An ablation on components and design choices of the proposed method is missing, although it's mentioned in the abstract. For example, how does GUM compare to LUM (Locally Unique Movement)?

**Summary Of The Paper:**

This paper investigated pruning for generative language models. They compared several existing pruning methods on decoder-only language models which had not been empirically evaluated before. The authors found that popular pruning methods such as movement pruning do not perform robustly on causal language models. In contrast, random pruning was shown to be a strong baseline. The authors propose two metrics to analyze the effectiveness of pruning. Based on these insights, they propose a new pruning method (GUM) which considers uniqueness of neurons. They evaluated this method on three benchmarks and showed some improvement over the prior methods.

**Summary Of The Review:**

This paper studies an important problem and provides some novel insights. The analysis and empirical evaluation part is mostly well executed except that the evaluation tasks/datasets could be expanded to improve representativeness and coverage. The effectiveness of the proposed method is not very clear with current results due to 1) improvements are small and may not be statistically significant. 2) the proposed method has high complexity which may slow down training and consume more memory. More clarity on both and some justification on whether the improvement outweigh the complexity would make it a strong paper.

---

> ### Author Response · Authors · 2022-11-08
> **(1/2) Addressing Concerns, Rerunning E2E, and Emphasizing our Main Claims**
>
>
>
> We thank the reviewer for their time in giving us feedback and asking very insightful questions, which will make our work stronger. Please see our responses below (1/2 comments, as we reached character limit).
>
> >The evaluation tasks and datasets do not have full coverage of how generative LLM are often evaluated. The authors evaluated on three tasks: language modeling (wikitext-103), text-to-code generation (wikiSQL), text-to-text generation (E2E NLG challenge). There are a couple of problems. First, E2E NLG challenge is not very diagnostic as the authors found that "E2E NLG challenge is not very diagnostic " and "speculate these discrepancies are due to the open-endedness of the problem domain and underfitting the data." I would expect the authors to evaluate on a more robust text-to-text task (e.g. summarization) to derive more informative evaluation results. Second, besides language modeling and text-to-code generation, LLMs are often used for language understanding and reasoning tasks, e.g. those evaluated in the GPT-3 paper. Experiments on pruning should be conducted on some of those tasks.
>
>
> Per the reviews, we have redone E2E results for GPT-2-sm, the change being a longer runtime for all methods, and note much more consistent results with the rest of our work. Please look for redone results for GPT-Neo in a future revision.
>
> We agree that including another experiment would improve the quality and message of the paper. We will work quickly to include 1 more experiment on a language reasoning or understanding task by the end of the review period, though we are not fully confident we will have time to produce this.
>
> However, we'd like to emphasize that we cannot find any other structured pruning results for generative models on a dataset like E2E. Therefore, the evaluation results are in fact a novel point themselves. Measuring performance on E2E can be difficult given the subjective nature of the problem.  We believe that our results on wikitext and wikisql show the necessary results to support the claims we make in our work, which is that both uniqueness and sensitivity should be considered for structured pruning of generative models.
>
> >The proposed approach computes cosine similarity for any two neurons' outputs. Although it's a sound solution, it is computationally expensive. Also, I wonder whether the running cosine similarity has high variance and how it affects the effectiveness of pruning.
>
>
> You have mentioned a very important concern about computation cost of the methods. In fact, it turns out computing (running) cosine similarity is relatively inexpensive compared to any kind of movement, given movement involves sorting all mask scores. Since GUM is built on top of movement, the full runtime of GUM is typically only around 1.1x slower at most versus regular hard movement, however, both are very significantly slower than no pruning (full-finetuning). We regretfully did not include this in the original draft, but we will certainly update accordingly soon.
>
> In addition, yes, it is a good point that cosine similarity is certainly quite noisy per-batch. However, our "running cosine" operation (Algorithm 1) also addresses this problem.
>
> >Improvements from GUM are minor especially on language modeling.
>
>
> Please see our above, separate comment directed at all reviewers regarding the novelty in our work. We acknowledge this limitation, however, the superiority of GUM is not the prime focus in this work. Our main message is to investigate traditional structured pruning of generative models, which (to our knowledge) has not been done before. We find many interesting trends, especially regarding the current best methods, and create an analysis framework based on these results. GUM is therefore a first attempt at working within this framework, and our results show its efficacy. However, we do not think it's the be-all and end-all pruning method.

---

> ### Author Response · Authors · 2022-11-08
> **(2/2) Continued Responses to Concerns from Reviewer**
>
> Please see our continued responses below, as we ran out of character limit.
>
> >An ablation on components and design choices of the proposed method is missing, although it's mentioned in the abstract. For example, how does GUM compare to LUM (Locally Unique Movement)?
>
>
> We had a slight issue in uploading the work - on accident, supplementary material was cut off of the main work. We provide ablations in the appendix (uploaded previously separately in supplementary material, fixed in latest revision). However, we will also aim to provide more comparisons with LUM in a draft by the end of the review period.
>
> >The authors could provide more motivation around why pruning is only performed on MLP layers.
>
> For generative models only, highly optimized inferencing code is able to cache attention states while generating new tokens. Therefore, MLP layers take a disproportionate amount of time when inferencing new tokens, especially for long sequences, which greatly matters in practical use. In addition, it is a common belief that MLPs hold "factual knowledge" in generative models, making them particularly difficult to prune. We have provided more of this motivation on the second page in this revision.
>
> >Different from other methods, GUM does not benefit from combining with distillation on language modeling (Table 4 & 5). Any insights on why?
>
>
> This is indeed curious, however, we note that this trend holds for all models at 75\% leftover, and sometimes for other methods at other percentages. Distillation can be slightly inconsistent. We speculate it could relate to using PPL as the evaluating metric rather than something like accuracy, given it depends on the model confidence directly. There are also some claims distillation is not always useful for generalization (i.e. https://proceedings.neurips.cc/paper/2021/hash/376c6b9ff3bedbbea56751a84fffc10c-Abstract.html), which we could more directly acknowledge in the paper if the reviewer finds useful.
>
> >Could you provide comparison on training speed (GUM vs. other methods)?
>
>
> Mentioned above - we will provide precise numbers, but we will mention now the difference is extremely small, as the TopK operation itself is highly costly (given it involves sorting scores of all neurons). Soft movement is very significantly worse than other methods (around 2-3x). We will add this information in the next revision.
>
> > The improvement from GUM are small in most cases. Are they statistically significant? Could you provide results from multiple runs with different seeds and show the variance?
>
>
> This is a good point - we will work to run multiple seeds on all experiments by the end of the review period, though we note that this many experiments requires a long runtime to complete. We will first prioritize a new dataset as you and others have suggested.

---

### Official Review · Reviewer_jU8R · 2022-10-22

**Confidence:** 4
**Correctness:** 4
**Technical Novelty And Significance:** 3
**Empirical Novelty And Significance:** 3
**Recommendation:** 6

**Clarity, Quality, Novelty And Reproducibility:**

The paper very clearly explains the main idea, methods and experiments. The experiments are well-chosen and executed, i.e. of good quality. There are original aspects of the work, e.g. the focus on sensitivity and uniqueness.

**Strength And Weaknesses:**

Strengths:

1. The topic of compressing LLM for generation tasks is an important one and I am happy to see that the authors focused on it.

2. The introduced sensitivity and uniqueness measures will be useful to future work in the area.

3. The experiments are thorough and demonstrate meaningful trends.

Weaknesses:

1. There are marginal improvements and not all of them are consistent. That is OK, but it probably signals that better methods than GUM can be constructed. I am curious to learn what the opinion of the authors is.

2. What happens if we use these pruning methods on non-generative tasks? I think it would be good to see that experiment in the paper, because the GUM method, by construction, is agnostic to the type of downstream task.

Minor:

* The results for the E2E NLG Challenge are not linked in the paragraph in the main text. Please link to Table 7.

**Summary Of The Paper:**

The authors conduct a study of structured pruning methods on natural language generation tasks. They demonstrate that the prior art do not improve significantly over the naive random pruning baseline. Then the authors attempt to understand the results through two measures that they introduce: sensitivity and uniqueness. The insights from these two measures allow the authors to construct a new pruning method, GUM, that explicitly enforces uniqueness through a cosine similarity proxy of the measure. The methods are implemented on large GPT-like models on a collection of downstream tasks.

**Summary Of The Review:**

There are useful contributions in this work, which I think will benefit the community. The work is very well-presented too. The topic of pruning for natural language generation is an important one, in my opinion, so I recommend weak acceptance.

---

> ### Author Response · Authors · 2022-11-08
> **Thanks, and Responses to Reviewer Concerns**
>
>
>
> We would like to thank the reviewer for the review and questions. We have responses to points raised by the reviewer below:
>
>
> >There are marginal improvements and not all of them are consistent. That is OK, but it probably signals that better methods than GUM can be constructed. I am curious to learn what the opinion of the authors is.
>
> We agree and genuinely hope that better methods than GUM can be constructed, and we believe our analysis for pruning generative models will likely shed light to this direction. As in our comment above, GUM serves as a proof-of-concept method to prove the effectiveness of our analysis framework, as its performance gains correlates with the improvements of its redundancy measures. Our framework also signals the potential direction for improvements, for example, we believe if there are methods that can push the boundary on both sensitivity and uniqueness, they would be likely to outperform all current methods. However, we also point out that under distillation, gaps between methods would likely close and the superiority of certain methods will fade, which is also the reason why our methods are not improving much when distillation is applied.
>
> > What happens if we use these pruning methods on non-generative tasks? I think it would be good to see that experiment in the paper, because the GUM method, by construction, is agnostic to the type of downstream task.
>
>
> This is correct, GUM would easily transfer by design. However, our scope is limited specifically to generative model, as our research goal is to investigate generative pruning in a systematic manner, which is lacking in the literature, and all our analysis are confined to it. We are also curious about the performance of GUM method in other domain, however we hope to investigate other domain systematically, which would be left as a future work.

---

> ### Author Response · Authors · 2022-11-18
> **New Dataset Added + Changes**
>
> Hello,
>
> Per recommendation from the reviewers, we have worked to include results from another dataset, SAMsum. This dataset is a text to text summarization dataset, which we find challenging for generative models. For this dataset, the trend follows that of other datasets and GUM mostly outperforms TopK/Hard Movement.
>
> We continued to run experiments on E2E and found still many inconsistencies in performance, even after further hyperparameter searching. We believe this dataset is simply not right for our purposes and thank the reviewers for their feedback to improve our paper. E2E has been moved to the appendix, for posterity and to show others who might be interested in using this dataset for pruning experiments.
>
> Please note, this latest revision also includes a few other updates per reviewer comments, including runtimes for the methods which show GUM is not much slower than Hard Movement, while Soft Movement is disastrously slow.
>
> Please let us know if you have any additional feedback which would help improve our paper further, or if you have any further questions.

---

### Official Review · Reviewer_AWKv · 2022-10-23

**Confidence:** 3
**Correctness:** 3
**Technical Novelty And Significance:** 3
**Empirical Novelty And Significance:** 2
**Recommendation:** 5

**Clarity, Quality, Novelty And Reproducibility:**

The paper is easy to follow. The proposed measures and GUM are intuitive. However, the paper didn't provide an explanation of why the soft movement method scores poorly on uniqueness and saliency. It's also unclear why the GUM only has regularization on uniqueness, but not saliency. The lack of sensitivity seems to be a more severe problem (much lower than the uniqueness) according to figures 1 and 2.

**Strength And Weaknesses:**

Strength:
1. The paper conducted a systematic evaluation of several structured pruning methods on the generative language model.
2. The proposed redundancy measures are intuitive and are shown to have a strong correlation with the pruning results.
3. The newly proposed GUM achieves strong performance when compared to the baseline methods.

Weaknesses
1. Most of the experiments focus on the WikiSQL and Wikitext tasks. More experiments and diverse tasks are needed to prove that the GUM can generalize to different tasks.
2. The result of E2E NLG is confusing. Although the author provided some explanation, it's still unclear what point the experiment wants to make. If the E2E NLG is not a good task to test the method, another text-to-text task should be used.

**Summary Of The Paper:**

The paper systematically studied the task of structured pruning of generative language models. It tests the performance of several different pruning methods on GPT-2 and GPT-Neo. The author also proposes two redundancy measures for each neuron in the MLP layer and shows that there is a strong correlation between these measures and the performance of pruning methods. Based on the observation, the author proposes a new pruning method, named GUM. The method combines a uniqueness regularization and a global top_v strategy to achieve strong performance on several pruning tasks.

**Summary Of The Review:**

Overall the paper proposes two interesting redundancy measures for pruning and a promising method GUM. But the current experiment section can be further improved and more analysis on the movement methods can be done.

---

> ### Author Response · Authors · 2022-11-08
> **Emphasizing Novelty, and Responses to Reviewer Concerns**
>
> We thank the reviewer for their time and feedback. Please see our responses to your raised points below.
>
> >Most of the experiments focus on the WikiSQL and Wikitext tasks. More experiments and diverse tasks are needed to prove that the GUM can generalize to different tasks.
>
> We appreciate your feedback on the lack of diverse tasks we tried. However, even on these two tasks, our experiments has shown consistent correlations between the two redundancy measures and the performances of multiple prevalent methods. The goal of these methods is **not to promote GUM as a much superior method than the others** (especially under distillation), but to **support our claim that both uniqueness and sensitivity are valuable properties that pruning methods should incorporate**. We believe the consistent redundancy patterns shown by these methods and correlating performance demonstrate the value of our work, with potential to improve the results.
>
> >The result of E2E NLG is confusing. Although the author provided some explanation, it's still unclear what point the experiment wants to make. If the E2E NLG is not a good task to test the method, another text-to-text task should be used.
>
> Per the reviews, we have redone E2E results for GPT-2-sm, the change being using a much longer number of training step for both finetuning and pruning, and note much more consistent results with the rest of our work. Please look for redone results for GPT-Neo in a future revision.
>
> We'd also like to emphasize that we cannot find published results on such a dataset previously, making this a novel finding in itself. The message with including E2E is to show that measuring the performance of generative models can generally be a difficult domain to work in as "quality" can be subjective. Therefore, this experiment is less about showing off GUM and more about testing pruning on these models in general.

---

> ### Author Response · Authors · 2022-11-08
> **Additional Responses to reviewer concerns**
>
> Please see our further responses below:
>
> > The paper is easy to follow. The proposed measures and GUM are intuitive. However, the paper didn't provide an explanation of why the soft movement method scores poorly on uniqueness and saliency.
>
> We explain the poor performance of soft movement in section 3.1, but to elaborate, soft movement largely struggles due to its extreme runtime and highly sensitive hyperparameters. We ran many experiments testing soft movement, but these experiments take 2-3x longer than the already lengthy hard movement experiments (around 30-40 hours of 8x V100 GPUs). In an ideal world where we can instantly test all hyperparameters, soft movement might perform well. However, we see this extremely long runtime to make the method impractical, especially when large language models already consume so many resources. If this is unclear, we may update the paper in the next revision to provide more discussion.
>
> > It's also unclear why the GUM only has regularization on uniqueness, but not saliency. The lack of sensitivity seems to be a more severe problem (much lower than the uniqueness) according to figures 1 and 2.
>
> It would not be congruent to regularize sensitivity because GUM is built on TopK, which can already be seen as an exact solution to finding the parameters with the highest sensitivity. Top-K here represents choosing the top k% of highest sensitivity parameters.
>
> Sensitivity in figures 1 and 2 is measured as a percentage of the finetuned model. Because sensitivity is a strictly positive value and pruned models will have less parameters than the finetuned model by definition, it is extremely likely that pruned models will have lower sensitivity than the finetuned model.
>
> Sensitivity indeed decreases dramatically, but in fact, TopK already does well here. With only 10% of parameters, TopK maintains around 40% of the model's sensitivity (in figure 1) - this is no small feat.
>
> Therefore, there is not a reason to further modify movement in terms of sensitivity, or at least, uniqueness is a much more pressing issue, as TopK does not have any built-in mechanisms to target uniqueness.

---

> ### Author Response · Authors · 2022-11-18
> **New Dataset Added**
>
> Hello,
>
> Per recommendation from the reviewers, we have worked to include results from another dataset, SAMsum. This dataset is a text to text summarization dataset, which we find challenging for generative models. For this dataset, the trend follows that of other datasets and GUM mostly outperforms TopK/Hard Movement.
>
> We continued to run experiments on E2E and found still many inconsistencies in performance, even after further hyperparameter searching. We believe this dataset is simply not right for our purposes and thank the reviewers for their feedback to improve our paper. E2E has been moved to the appendix, for posterity and to show others who might be interested in using this dataset for pruning experiments.
>
> Please note, this latest revision also includes a few other updates per reviewer comments, including runtimes for the methods which show GUM is not much slower than Hard Movement, while Soft Movement is disastrously slow.
>
> If you find our revision to address some of your concerns, we ask to let us know or also further discuss if you have any additional concerns.

---

### Official Review · Reviewer_MVrZ · 2022-10-25

**Confidence:** 5
**Correctness:** 3
**Technical Novelty And Significance:** 3
**Empirical Novelty And Significance:** 3
**Recommendation:** 6

**Clarity, Quality, Novelty And Reproducibility:**

- Presentation suggestions:
  - Does Top_v throughout the presentation refer to Hard movement pruning in Table 2? If so, it would help the reader to uniformize these short names.
  - I have found not intuitive to present results with GUM (Figure 1 and 2) before introducing GUM
- Why is the sensitivity decreasing as the model is pruned? I would have expect that the remaining weights have a greater impact on the final output and thus as you prune, the average sensitivity increases.
- Section 3.2: you mention a few times that poor sensitivity and uniqueness explains poor benchmark performance. Is it a causality link? A correlation? Could you articulate the intuition behind?
Examples:
  - *“Magnitude pruning universally scores worst on both metrics, explaining its poorer performance in all experiments”*
  - *“which partially explains why distillation improves it significantly”*
  - *“explaining its superiority across various tasks”*
- What is the maximum number of epochs you tried for block soft movement pruning? In my experience, this method requires a very slow pruning (5x to 10x more steps than hard weight movement pruning).
- What are the trends for extreme pruning (i.e. less than 10% leftover) for Table 2, 3 and 4, 5? Does it lead to more unequivocal results?
- *“Random pruning obtains similar distillation sensitivity and uniqueness, though slightly lower, to hard movement, lending credence to its overall high performance. However, sensitivity is markedly lower without distillation as is reflected in all tables. We point to this as proof that hard movement does not target uniqueness.”* Could you expand on this insight? I have found it to be a very generous conclusion from Figure 1 and 2. Handholding the reader for that conclusion might bring some clarity.


**Strength And Weaknesses:**

Strengths:
- The problem is relatively well-motivated and the paper makes a noticeable effort to be didactic.
- The contributions are somewhat novel, and experiments are well conducted on reasonably large setups.
- The result about gradual pruning performing similarly to previous state-of-the-art structured pruning is surprising and insightful.

Weaknesses:
- The numbers showing the superiority of the method are weak or show only weak improved performance or trends.
- The connection between sensitivity/uniqueness and fine-pruning performance is not well articulated (see questions).
- I have doubts about whether the choice of benchmarks is the most appropriate. For instance, the authors note that on E2E, pruned models perform better than the non-pruned baseline.


**Summary Of The Paper:**

The authors are interested in structured pruning of generative language models. In particular, this work builds on top of [Movement Pruning](https://arxiv.org/abs/2005.07683) and [Block movement pruning](https://arxiv.org/abs/2109.04838) to prune entire structures (and not individual weights) in models similar to GPT2 and the fine-tuning stage.

The authors first notice that current structure pruning applied to decoder only language models on NLG tasks perform relatively similarly at pruning rates between 10% and 50% (percentage of remaining weights), and more surprisingly, similarly to random pruning.

The authors then introduce two fundamental measures of redundancy called “sensitivity” and “uniqueness” which respectively measure how much a group of parameters impact the training objective and how unique a group of parameters is compared to other groups of parameters. This motivates the introduction of Globally Unique Movement, a method that essentially encourages remaining neurons to be dissimilar (as measured by cosine similarity), by modifying the regularization in movement pruning.


**Summary Of The Review:**

This paper extends movement pruning and block movement pruning by introducing a "uniqueness" metric in the training objective which encourages the model to prune similar neurons.
While the method and insights are relatively novel, the numbers and comparisons of previous pruning methods are somewhat weak.

---

> ### Author Response · Authors · 2022-11-08
> **(2/2) Responses to concerns from the reviewer**
>
>
>
> >What is the maximum number of epochs you tried for block soft movement pruning? In my experience, this method requires a very slow pruning (5x to 10x more steps than hard weight movement pruning).
>
> One large problem we found with soft movement was its extreme runtime (sec 3.1). As other reviewers have mentioned, we did not include real runtime numbers, which we will include in the next revision.
>
> To summarize, scanning hyperparameters with 5x to 10x more steps for soft movement would be incredibly unwieldy, on the order of many hundreds of hours per single run in our case. We acknowledge it is possible this would lead to soft movement beating hard movement  (still, our gut feeling is that if hard movement was also given the same number of training steps, it would still outperform), however, this makes the method entirely impractical to use for realistic hyperparameter scanning.
>
> Epochs are listed in the appendix (provided separately in supplementary materials - we had a slight issue with the website when uploading this with the main work, fixed in current revision), but we believe soft movement should have had time to perform well.
>
> >What are the trends for extreme pruning (i.e. less than 10\% leftover) for Table 2, 3 and 4, 5? Does it lead to more unequivocal results?
>
> We struggled to choose exact pruning percentages - applied readers would likely care about maximum \% removed with negligible performance loss, while theoretical readers would likely care about such extreme pruning cases. This is a good question, but we likely cannot answer definitively before the end of the review period. If we were to speculate, the answer is likely yes.
>
> >"Random pruning obtains similar distillation sensitivity and uniqueness, though slightly lower, to hard movement, lending credence to its overall high performance. [...]  Handholding the reader for that conclusion might bring some clarity.
>
>
> There is no obvious built-in metric or mechanism to target maximum uniqueness in hard movement - it's purely a sensitivity-based method. Our claim here is that we can measure this effect empirically as shown in Figures 1 and 2, as the uniqueness of hard movement is about the same as that of random pruning. Since random pruning surely does not target uniqueness more than any amount of pruning targets uniqueness, neither does hard movement. However, random pruning performs remarkably well when used with distillation, and we see that it obtains high sensitivity scores in this case. Under distillation, sensitivity seemingly converges for most methods, which is perhaps an interesting phenomenon on its own.
>
> We did make a slight error here - your quoted text should say figures, not tables, which may explain some confusion. This is a mistake, and we have fixed it in the current revision.

---

> > ### Comment · Reviewer_MVrZ · 2022-12-04
> > **thank you for your response**
> >
> > Thank you so much for your response.
> > I really appreciate that you take into consideration all the reviewers' comments and suggestions and implemented some of them.
> > In particular, the addition of the Samsum numbers is appreciated.
> > At this point, I am not changing my score as I believe the weaknesses I pointed out in the initial review (the connection between pruning performance and uniqueness/sensitiveness is articulated as a correlation, and I don't know to what extent this is an actionable observation + failure to get stronger numbers).

---

> ### Author Response · Authors · 2022-11-08
> **(1/2) Responses to Concerns from the reviewer**
>
>
> We thank the reviewer for their in-depth, fair review and insightful comments. We would like to respond the concerns raised by the reviewer. Please see our responses below (1/2 as we ran out of character limit, additional comment below):
>
> >The numbers showing the superiority of the method are weak or show only weak improved performance or trends.
>
> Please see our above comment concerning the novelty in our work. Overall, our main goal in this work is work is to provide an initial investigation into structured pruning for generative models, as literature is sorely lacking in this area, not to promote GUM as a superior method than all others. GUM is largely a proof-of-concept method to show the validity of our investigation and analysis, rather than a one-size-fits-all pruning method, and therefore relatively small performance gains are acceptable for our purpose. We believe our analysis will provide grounds for further development in the area.
>
> >The connection between sensitivity/uniqueness and fine-pruning performance is not well articulated (see questions).
>
> Please see responses below. Overall, we have provided more explanation and conclusions on how sensitivity and uniqueness relate to final fine-pruning performance in the current revision.
>
> >I have doubts about whether the choice of benchmarks is the most appropriate. For instance, the authors note that on E2E, pruned models perform better than the non-pruned baseline.
>
> Per the reviews, we have redone E2E results for GPT-2-sm, the change being using a much longer number of training step for both finetuning and pruning, and note much more consistent results with the rest of our work. Please look for redone results for GPT-Neo in a future revision.
>
> Our goal with including E2E was to show inconsistency in measuring generative model performance, as evaluating the "quality" of writing can be a difficult task. We'd like to again emphasize that we have not found any studies examining structured pruning performance for generative models on such a task, so in fact, the messiness of the results is a novel point in itself. This suits our main purpose of the work, which is to take a look at "what matters when pruning generative language models".
>
> However, we will aim to include 1 more dataset by the end of the review period given the response from reviewers.
>
>
> >Does Top v throughout the presentation refer to Hard movement pruning in Table 2? If so, it would help the reader to uniformize these short names.
>
> Correct, Top-V is hard movement. We will clarify this further in the current revision.
>
> >I have found not intuitive to present results with GUM (Figure 1 and 2) before introducing GUM.
>
> Understood, we will consider moving these plots. However, this was partially done to de-emphasize the importance of GUM, and greater emphasize the importance of our analysis in general. Please let us know what you think, and if this causes major detriment to the work.
>
> >Why is the sensitivity decreasing as the model is pruned? I would have expect that the remaining weights have a greater impact on the final output and thus as you prune, the average sensitivity increases.
>
> Sensitivity here is measured as a sum of all sensitivities from neurons, as a percentage of the original model - not the average sensitivity. Sensitivity is a strictly positive value, so fewer weights will very likely give less total sensitivity compared to baseline. You are correct that average sensitivity should increase, and we will attempt to include this plot as well. We have made this more clear in the current revision.
>
> >Section 3.2: you mention a few times that poor sensitivity and uniqueness explains poor benchmark performance. Is it a causality link? A correlation? Could you articulate the intuition behind? Examples [...]
>
> This is a very good point. Ultimately, this is a correlative link, rather than causal. We cannot have counter-factual analysis to know whether it is causal, given we cannot directly manipulate the sensitivity and uniqueness of the model alone. We see our analysis framework as a helpful tool to design pruning methods, and the overall better performance of GUM shows that uniqueness indeed matters.
>
> Our message is to demonstrate that worse methods typically score lower on both metrics, and better methods typically score higher on both metrics. When designing a new pruning method to test the validity of our pruning framework, we find performance increases overall. In the current revision, we directly acknowledge this limitation, and provide more discussion.

---

> ### Author Response · Authors · 2022-11-18
> **New Dataset Added**
>
> Hello,
>
> Per recommendation from the reviewers, we have worked to include results from another dataset, SAMsum. This dataset is a text to text summarization dataset, which we find challenging for generative models. For this dataset, the trend follows that of other datasets and GUM mostly outperforms TopK/Hard Movement.
>
> We continued to run experiments on E2E and found still many inconsistencies in performance, even after further hyperparameter searching. We believe this dataset is simply not right for our purposes and thank the reviewers for their feedback to improve our paper. E2E has been moved to the appendix, for posterity and to show others who might be interested in using this dataset for pruning experiments.
>
> Please note, this latest revision also includes a few other updates per reviewer comments, including runtimes for the methods which show GUM is not much slower than Hard Movement, while Soft Movement is disastrously slow.
>
> If you find our revision to address some of your concerns, we ask to let us know or also further discuss if you have any additional concerns.

---

### Author Response · Authors · 2022-11-08
**Emphasizing the Novelty in our Work**

To all reviewers,

We appreciate the reviewers' feedback and observations. We noticed that the reviews pointed out the modest improvement of GUM over other pruning methods and we'd like to clarify the contribution of our work for all reviewers.

Our work aims to explore structured pruning for generative models, which is a novel direction (as far as we know). Therefore, our work is mainly novel in providing the first structured pruning results on these models, explaining and analyzing these results, and then using our analysis to propose a better method.

We have empirically shown the difficulties of the pruning generative models and diagnosed these difficulties with our proposed framework. We observe that the (widely used or most popular) state-of-the-art method for encoder-decoder models, soft movement, fails to perform well for some reasons, and its variant, hard movement, only slightly outperforms random pruning. Our main focus is to understand why this happens, and we hypothesize that a successful structured pruning method needs to consider both uniqueness and sensitivity of neurons rather than only sensitivity.

The performance gain of GUM correlates with the improved balance of its redundancy measures, confirming the validity of our framework and analysis. GUM is a proof-of-concept that shows the usefulness of our framework. We validate this usefulness by evaluating the uniqueness and sensitivity of remaining neurons after pruning, and we find trends that support our hypothesis. We admit that this method has only a small advantage, but it still has a similar advantage to that of hard movement over random pruning. We believe this framework can inspire future pruning methods to further enhance performance and we've revised our text to emphasize this point.

---

### Decision · Program_Chairs · 2023-01-20

**Decision:**

Reject

**Justification For Why Not Higher Score:**

The complete lack of comparison against Wang et al. 2019 who tackle the exact same problem (i.e., structured pruning of generative LMs) makes it difficult to justify accepting this paper.

**Justification For Why Not Lower Score:**

N/A

**Metareview: Summary, Strengths And Weaknesses:**

This paper studies structured pruning (wherein neurons are pruned out) in generative models. The strength of the paper is the application to generative modeling, and a (somewhat) novel finding that random pruning works well. The main weakness is the lack of comparison against baselines other than movement pruning. For example Wang et al. 2019 (https://arxiv.org/abs/1910.04732) also study pruning of generative models using relaxations of discrete variables.

**Summary Of Ac-Reviewer Meeting:**

Some points raised by the reviewers were:

- The fact that the E2E benchmark was somewhat unusual, although it was partially addressed in the rebuttal.
- Movement pruning baseline, and more generally, lack of baselines against other structured pruning works such as Wang and Lei 2019 (https://arxiv.org/abs/1910.04732)
- The fact that the largest model they tested on was only a 1.3B model.
- The fact that this was only focusing on generative models, even though there is nothing special about the technique that is tailored for generative models.

The first point (i.e., E2E benchmark) seems to have been addressed in the rebuttal, but the other points were not.